# Short-Term Effect of a Low-Protein High-Carbohydrate Diet on Mature Female and Male, and Neomale Rainbow Trout

**DOI:** 10.3390/ijms22116149

**Published:** 2021-06-07

**Authors:** Nathan Favalier, Vincent Véron, Michael Marchand, Anne Surget, Patrick Maunas, Nicolas Turonnet, Stéphane Panserat, Lucie Marandel

**Affiliations:** NUMEA, INRA, University Pau & Pays Adour, E2S UPPA, 64310 Saint-Pée-sur-Nivelle, France; nathan.favalier@inrae.fr (N.F.); vincent.veron@inrae.fr (V.V.); michael.marchand@inrae.fr (M.M.); anne.surget@inrae.fr (A.S.); patrick.maunas@inrae.fr (P.M.); nicolas.turonnet@inrae.fr (N.T.); stephane.panserat@inrae.fr (S.P.)

**Keywords:** broodstock, glucose metabolism, lipid metabolism

## Abstract

Rainbow trout are considered as a poor user of dietary carbohydrates, displaying persistent postprandial hyperglycaemia when fed a diet containing high amounts of carbohydrates. While this phenotype is well-described in juveniles, less attention was given to broodstock. Our objective was to assess for the first time the short-term consequences of feeding mature female and male, and neomale trout with a low-protein high-carbohydrate diet on glucose and lipid metabolism. Fish were fed for two days with a diet containing either no or 32% of carbohydrates. We analysed plasma metabolites, mRNA levels and enzymatic activities of glycolysis, gluconeogenesis, de novo lipogenesis and β-oxidation in the liver. Results demonstrated that the glucose and lipid metabolism were regulated by the nutritional status in all sexes, irrespective of the carbohydrate intake. These data point out that carbohydrate intake during a short period (5 meals) at 8 °C did not induce specific metabolic changes in broodstock. Finally, we demonstrated, for the first time, sex differences regarding the consequences of two days of feeding on glucose and lipid metabolism.

## 1. Introduction

Salmonid species, and more specifically rainbow trout (*Oncorhynchus mykiss*), are carnivorous species and are poor users of digestible carbohydrates. Nonetheless, the inclusion of carbohydrates in the aquafeed formula stems from the replacement of fish-derived products in recent years by more sustainable ingredients, such as animal processed proteins and plant-derived products [1]. However, when the substitution of fish meal (FM) by digestible carbohydrates exceeds 20% in the aquafeed formula, in the long term, rainbow trout display a decrease of growth and a persistent postprandial hyperglycaemia [2,3,4]. Several hypotheses have been proposed to explain this phenotype. Among them is the non-inhibition of the production of endogenous glucose by the liver through gluconeogenesis [5]. De novo lipogenesis is another pathway proposed to be involved in the above phenotype through glucose storing by the conversion of glucose to free fatty acid [6]. The de novo lipogenesis is indeed poorly induced by digestible carbohydrates, but when this pathway is activated by anti-diabetic drugs (metformin), it leads to a better postprandial glycaemic control [6,7,8].

While glucose metabolism is well-described for juvenile rainbow trout [4,8], only few studies have focused on this metabolic pathway in broodstock [9,10,11,12,13] and described it during gametogenesis in males and females. However, only one study addressed questions around the effect of feeding broodstock with a diet enriched in carbohydrates, and specifically on reproductive performance by demonstrating higher fecundity and survival rate of progeny of females fed with a diet containing white flour compared to females fed with a controlled diet [10]. More recently, Callet et al. reported that female and male broodstock are able to eat and use a high-carbohydrate diet (35%) during the whole reproductive cycle for females (9 months) and during the early part of the cycle for males (4 months), as reflected by the growth performance and the success of the reproduction process. Both sexes were never hyperglycaemic when fed with a high-carbohydrate diet, to the contrary of what is commonly observed in juveniles and resulting from a particular regulation of both the glucose and the lipid metabolism [14]. However, it is not clear whether the modulation of the glucose metabolism, which may explain their ability to eat and grow under a high-carbohydrate diet compared to juveniles, is related to their physiological stage (broodstock) or results from a long-term consequence/adaption of the intake of digestible carbohydrates (4 and 9 months respectively, for males and females). Moreover, Callet et al. evaluated the long-term consequences of a high-carbohydrate diet only in female and male broodstock. However, a third sex, called neomale, is commonly used in aquaculture to produce all-female mono-sex populations [15]. To our knowledge, no study has investigated the glucose and lipid metabolism of these particular individuals fed a high-carbohydrate diet.

In this context, the present study aimed to assess the short-term consequences on glucose and lipid metabolism in mature females and males, as well as in neomales, when fed a high-carbohydrate diet. To do so, we fed fish for two days (5 meals) with a 32% carbohydrate diet or a control diet containing no carbohydrates. At the end of this trial, we analysed plasma metabolites, mRNA levels and activities of glucose and lipid metabolism-related actors. All these parameters were studied in the liver, which has been demonstrated as a key organ for the regulation of intermediary metabolism.

## 2. Results

### 2.1. Zootechnical Parameters

Mean body weight was measured for fasted, NC and HC females, males and neomales (Figure 1a). Differences were not statistically significant between fasted females and NC and HC females as well as between NC females and HC females. Similar results were also found for males and neomales. Concerning the feed intake, females displayed no differences between females fed with the NC diet and females fed with the HC diet with around 20 g of feed per fish. No significant differences were found in males or in neomales eating 35–38 and 38 g per fish and day, respectively.

### 2.2. Results in Females

#### 2.2.1. Plasma Glucose and Triglycerides Concentrations in Females

Plasma glucose and triglycerides concentrations were investigated for fasted females and females fed either the NC or the HC diet (Figure 2). The glucose plasma concentrations (Figure 2a) were not significantly different between females fasting 15 days and females fed with the NC or HC diet. Concerning the fed females, females with the NC diet had a significantly lower glycaemia compare to females fed the HC diet (*p* = 0.02). For triglycerides plasma concentrations (Figure 2b), no differences were statistically significant between fasted females and fed females as well as between NC and HC females.

#### 2.2.2. Relative mRNA Levels of Glucose and Lipid Metabolism-Related Genes in the Liver of Females

The mRNA levels for glucose transporter, glycolysis, gluconeogenesis, de novo lipogenesis and β-oxidation-related genes were assessed in the liver of fasted females and in females fed with either the NC or HC diet (Table 1). For the genes *glut2a* and *glut2b* involved in the transport of glucose, mRNA levels in fasted and fed females as well as NC and HC females were not statistically different. For the genes involved in the glycolysis, no differences were statistically significant between fasted females and females fed with the NC or HC diet. For fed females, only *gckb* displayed differences among the conditions, with a higher mRNA level for HC females compared to NC ones. Concerning the gluconeogenesis, only genes coding for the enzyme involved in the last step of this pathway (glucose-6-phosphatase) displayed differences in mRNA levels between our experimental conditions. *g6pca* displayed higher mRNA levels in fasted females compared to HC females. *g6pcb2a* exhibited lower mRNA level in fasted females compared to females fed with the HC diet. Finally, *g6pcb1a* and *g6pcb1b* exhibited higher mRNA levels in fasted females than in females fed with the HC diet. For the de novo lipogenesis, *srebf1*, *acly*, *aca-**α**a*, *aca-**α**b* and *fasn* exhibited higher mRNA levels for females fed with the NC or HC diet compared to fasted females.

#### 2.2.3. Enzymatic Activities of Glucose and Lipid Metabolisms in Liver of Females

Enzymatic activities were analysed and compared between females fasted for 15 days, females fed with the NC diet and females fed with the HC diet for 2 days. Concerning glycolysis enzymatic activities (Figure 3a), glucokinase (Gck) exhibited no significant differences between fasted females and females fed with the NC or HC diet. Phosphofructokinase (Pfk) exhibited a lower activity in liver of fasted females compared to females fed with the HC diet, and the difference was not statistically significant between NC and HC females. Concerning the pyruvate kinase (Pk), fasted females exhibited a lower enzymatic activity compared to females fed with the NC or the HC diet, but no statistically significant difference was highlighted between females fed the NC and the HC diet. Concerning the gluconeogenesis (Figure 3b), the glucose-6-phosphatase (G6pc) and the fructose-1,6-bisphosphatase (Fbp) displayed no significant statistical differences between fasted females and females fed with the NC or the HC diet, as well as between females fed with NC diet and females fed with the HC diet. Finally, concerning the de novo lipogenesis-related enzyme (Figure 3c), differences were not statistically significant between fasted females and females fed with the NC or the HC diet, as well as between NC-fed females and HC-fed females for the fatty acid synthase (Fasn).

### 2.3. Results in Males

#### 2.3.1. Plasma Glucose and Triglycerides Concentrations in Males

The plasma glucose and triglycerides concentrations were analysed and compared between males fasted for 15 days and males fed with either the NC or the HC diet (Figure 4a,b, respectively). Males exhibited higher glycaemia when fasted compared to males fed for two days with either the NC or the HC diet (Figure 4a), but no significant differences were observed between males fed with the NC diet and males fed with the HC diet. Concerning the plasma triglycerides concentration, fasted males and males fed with the NC diet displayed lower levels of plasma triglycerides than males fed with the HC diet.

#### 2.3.2. Relative mRNA Levels of Glucose and Lipid Metabolism in Liver of Males

Relative mRNA levels of genes involved in the transport of glucose, glycolysis, gluconeogenesis, de novo lipogenesis and β-oxidation were analysed and compared between males fasted for 15 days and males fed with the NC or the HC diets (Table 2). For the transport of glucose, *glut2a* and *glut2b* displayed no significant differences between fasted males and males fed with NC or the HC diet, as well as between males fed with the NC diet and males fed with the HC diet. For the glycolysis, only the mRNA level of *gcka* significantly differed, with higher mRNA levels in fasted males compared to fed males. For the gluconeogenesis, only *g6pcb1b* and *g6pcb2a* displayed significant differences of mRNA levels between experimental conditions, whereas mRNA levels from other analysed genes remained stables. *g6pcb1b* displayed no significant differences between fasted and fed males. A higher mRNA level of *g6pcb1b* was observed in males fed with the NC diet compared to males fed with the HC diet. *g6pcb2a* mRNA levels were significantly lower in fasted males compared to fed males and the difference was not statistically significant between the NC and the HC diet. Finally, for *pck1*, males fed with the NC diet displayed higher mRNA levels compared to fasted males or males fed with the HC diet. Concerning the de novo lipogenesis, *srebf1*, *aca-αa*, *aca-αb* and *fasn* displayed lower mRNA levels in fasted males compared to fed males, but no significant difference was shown between males fed with the NC or the HC diet. Finally, the β-oxidation-related gene *aca-βa* displayed higher mRNA levels in fasted males compared to fed males, but no significant difference was highlighted between NC- and HC-fed males.

#### 2.3.3. Enzymatic Activities of Glucose and Lipid Metabolism in Liver of Males

Enzymatic activities were analysed and compared between males fasted for 15 days, males fed with the NC diet and males fed with the HC diet for 2 days. For enzymatic activities related to the glycolysis (Figure 5a), Gck, Pfk and Pk displayed no differences between fasted and fed males, and differences were not statistically significant between males fed with the NC diet and males fed with the HC diet. Similar results were found for the activities of enzymes related to the gluconeogenesis (Figure 5b).

### 2.4. Results in Neomales

#### 2.4.1. Plasma Glucose and Triglycerides Concentrations in Neomales

Plasma glucose (Figure 6a) and plasma triglycerides (Figure 6b) concentrations were analysed and compared in fasted neomales and neomales fed with either the NC diet or the HC diet. Concerning the glucose plasma concentration, fasted neomales displayed lower glycaemia compared to neomales fed with the HC diet, and differences were not statistically significant between neomales fed with the NC diet and neomales fed with the HC diet. Concerning the level of plasma triglycerides, no significant differences were demonstrated between fasted and fed neomales, as well as between neomales fed with the NC diet and neomales fed with the HC diet.

#### 2.4.2. Relative mRNA Levels of Glucose and Lipid Metabolism in Liver of Neomales

The mRNA levels of key genes involved in the glucose transport, the glycolysis, the gluconeogenesis, the de novo lipogenesis and the β-oxidation were investigated and compared between neomales fasted for 15 days and neomales fed for two days with either the NC or the HC diet (Table 3). For the transport of glucose, either *glut2a* or *glut2b* displayed no significant differences between fasted or fed neomales. For the glycolysis, *gckb* displayed higher mRNA levels in neomales fed with the HC diet compared to fasted neomales and neomales fed with the NC diet. *pfkla* and *pfklb* displayed significantly lower mRNA levels in fasted neomales compared to neomales fed with the NC or the HC diet. Concerning the gluconeogenesis, *g6pb1a* displayed higher mRNA levels in fasted neomales compared to fed neomales, and differences were not statistically significant between neomales fed with the NC and neomales fed with the HC diet. *g6pcb1b* displayed lower mRNA levels in fasted neomales compared to neomales fed with the NC diet, and differences were not statistically significant between neomales fed with the NC diet and neomales fed with the HC diet. Fasted neomales displayed lower mRNA levels for *g6pcb2a* compared to fed fish and differences were not statistically significant between the two types of diet. Concerning the fructose-1,6-biphosphatase, *fbp1b2* displayed higher mRNA levels in fasted neomales compared to neomales fed with the HC diet, and differences were not statistically significant between NC and HC fed neomales. Finally, with the phosphoenolpyruvate kinase, only *pck1* displayed significant differences, with lower mRNA levels in fasted neomales compared to fed neomales. Concerning the de novo lipogenesis, investigated genes displayed significant differences among experimental conditions: *srebf1*, *g6pd*, *acly*, *aca-αa*, *aca-αb* and *fasn,* with lower mRNA levels in fasted neomales compared to fed neomales, and differences were not statistically significant between NC and HC neomales. *aca-βa* and *aca-βb* displayed higher mRNA levels in fasted neomales compared to neomales fed with the HC diet and neomales fed with the NC diet, respectively. For *aca-βa* and *aca-βb*, differences were not statistically significant between the NC and the HC diet.

#### 2.4.3. Enzymatic Activities of Glucose and Lipid Metabolism in Liver of Neomales

Enzymatic activities were analysed and compared between neomales fasted for 15 days and neomales fed with the NC diet or fed with the HC diet for 2 days (Figure 7). For the glycolysis (Figure 7a), only Pk displayed different levels of activity between fasted neomales and neomales fed with the HC diet, and statistical differences were not significant between the two types of diet. For the gluconeogenesis (Figure 7b), statistical differences were not significant between fasted neomales and neomales fed with either the NC or HC diet for both G6pc and Fbp. Moreover, G6pc and Fbp displayed no statistical differences between fasted neomales and neomales fed with the NC or the HC diet, as well as between neomales fed with the NC diet and neomales fed with the HC diet. Finally, for the de novo lipogenesis (Figure 7c), Fasn displayed no statistical differences.

## 3. Discussion

Rainbow trout is a carnivorous fish metabolically adapted for high-protein catabolism and low-carbohydrate use, displaying persistent postprandial hyperglycaemia when fed a diet containing more than 20% of digestible carbohydrates [8]. It appears relevant to test, for the first time, the direct effects of feeding broodstocks with a high-carbohydrate diet. Indeed, broodstock are large animals consuming high amounts of feed. Mature females and males, and neomales, were fed with a non-carbohydrate diet or a high-carbohydrate diet for two days (5 meals), and glucose and lipid metabolism at the molecular level were investigated in the liver, the centre of intermediary metabolism regulation. First, we will discuss the molecular consequences of feeding broodstocks on their glucose metabolism to pursue on that of lipids with the de novo lipogenesis and the β-oxidation. Finally, we will conclude with the differences that we highlighted between all sexes studied.

### 3.1. Glucose Metabolism Is Regulated at the Molecular Level by the Nutritional Status, Irrespective of the Carbohydrate Intake

Feeding mature females and males, and neomales, has been effective regarding the regulation of glucose metabolism at the molecular level, and irrespective of the diet. Our results showed that feeding mature females and immature neomales for two days influenced the mRNA levels of glucose metabolism-related genes. Indeed, in females and neomales, we observed an overall upregulation of the glycolysis pathway. However, for fed females and neomales, we highlighted a poor regulation of the gluconeogenesis pathway by the nutritional status (except for *g6pc* genes) for all sexes investigated. Such regulation of the global glucose metabolism by the nutritional status mimics regulations previously reported in juvenile trout fasted and refed during a short period [5]. The maintenance of an active gluconeogenic pathway in fish fed a NC diet suggests a need for endogenous glucose production to sustain maintenance of normoglycaemia and provide glucose as fuel for physiological basal functions (such as brain or heart functioning) [16,17,18,19,20]. Furthermore, this is consistent with previous in vivo and in vitro studies showing that the gluconeogenesis flux is never turned off in fed carnivorous teleosts [21,22].

Except for the gluconeogenic *g6pcb1b* gene, which was downregulated in HC-fed trout, no significant differences were induced by the diet composition regarding the glucose metabolism. The absence of a drastic induction of glucokinase under the HC diet, which is a feature observed in rainbow trout fed such a diet [23], as well as the relatively low glycaemia observed, are in favour of a low assimilation of the dietary carbohydrates. This may be linked to a poor digestion of the carbohydrates contained in the HC diet due to the short time of feeding (2 days, 5 meals) and the low water temperature (8 °C) at which trout broodstock are usually reared [24]. However, we cannot exclude that the low glycaemia could also be linked to the low levels of lipids in the diets, amounts which sustain trout broodstocks’ needs. Indeed, low levels of lipids can be associated with a better glycaemia homeostasis in trout fed with carbohydrates [25]. All together, these data indicate that our experimental conditions, especially 5 meals, were not enough to trigger differences between the NC and HC diets. 

### 3.2. At the Molecular Level, the De Novo Lipogenesis Is Upregulated and the β-Oxidation Is Downregulated by Feeding, Irrespective of the Carbohydrate Intake

The lipid metabolism is closely linked to the fate of glucose in cells by its conversion to triglycerides through the de novo lipogenesis (DNL). Moreover, DNL is hypothesised to be involved in the glucose-intolerant phenotype of the rainbow trout. Indeed, when this pathway is induced by anti-diabetic medication (metformin) in trout, it leads to a better postprandial glycaemic control [6]. In this study, we were able to demonstrate that DNL is upregulated by the nutritional status regardless of the sex and irrespective of the diet. Coupled with the upregulation of the DNL, we also observed, in males and neomales, the attenuation of the β-oxidation. This is in accordance with what is commonly observed, as the DNL and β-oxidation are known to be regulated by several factors, including the nutritional status [26,27]. As for the glucose metabolism, we did not demonstrate differences between the NC and the HC diets regarding the lipid metabolism. However, an increased hepatic lipogenic pathway has been demonstrated after 10 weeks of feeding in rainbow trout fed with a high-carbohydrate diet compared to trout fed with a non-carbohydrate diet [28]. As mentioned above, this latter result supports the hypothesis that the duration of the feeding trial was too short to allow a sufficient digestion of dietary carbohydrates able to induce metabolic changes. 

### 3.3. The Two-Day Feeding of Mature Females and Males, and Neomales, Induced Different Regulation of the Glucose and Lipid Metabolism

In this study, we were able to demonstrate differences among sex regarding the glucose and lipid metabolism. First, we observed a lower feed intake in females compared to males and neomales. This can be explained by the fact that females eat less before and after the period of reproduction [29]. Indeed, reduction or complete cessation of feeding can be observed from a few days to several months during the reproduction in wild salmonids, and fasting is thus recommended for broodstock management (i.e., to avoid faeces or urine mixed with eggs) during reproduction [30,31]. This could explain the lower induction of DNL in fed females compared to males and neomales and the absence of attenuation of the β-oxidation. 

One of the major differences between sex observed in this study is the absence of the induction of hepatic glycolysis in males fed with the HC diet. This could be linked to the fact that males were milted during the sampling events and that the glucose uptake may be preferentially used to sustain testes requirements [12]. 

Finally, we were able to highlight more differences in nutritional regulation of gene expression in neomales, irrespective of the metabolism investigated. Neomales are sex-reversed females used to produce an all-female population, and thus they are kept until maturation and sacrificed to collect gametes [32]. The more pronounced modifications of their metabolism in response to nutritional status or HC diet (more particularly, the huge increase in *gcka* mRNA levels) compared to mature females or males is probably linked to their physiological status and weight close to juveniles at the time point of the experiment, which imposes on them a lower response latency. However, they did not display hyperglycaemia nor induction of *g6pcb2* genes when fed the HC diet as previously described in juveniles [5]. This could again be due to the hypothesised low assimilation of dietary carbohydrates in our experiment. However, regarding only the results obtained here, we cannot rule out that neomales may be able to better regulate their glucose metabolism than juveniles when fed a high-carbohydrate diet. Further studies are needed to perfectly describe the glucose and lipid metabolism of these particular broodstock, especially for longer periods of feeding during gametogenesis, as to our knowledge, no studies have focused on this point yet.

## 4. Materials and Methods

### 4.1. Fish Diet

Two experimental feeds for rainbow trout, i.e., NC (high-protein and low-carbohydrate diet) and HC (low-protein and high-carbohydrate diet), were prepared in our own facilities (INRAE, Donzacq, Landes, France) as extruded pellets. Gelatinised starch was included as the carbohydrate source, protein originated from fish meal and dietary lipids from fish oil and fish meal (Table 4). The two diets were isolipidic and isoenergetic. The amount of carbohydrate (~32%) was balanced by of the amount of protein. Chemical composition of diets was analysed through several steps: dry matter was determined after drying to constant mass at 105 °C, crude lipids were assessed by petroleum ether extraction (Soxtherm), crude proteins were assessed by the Kjeldahl method after acid digestion, gross energy was measured in an adiabatic bomb calorimeter (IKA, Heitersheim Gribeimer, Germany) and ash was estimated through incineration in a muffle furnace for 6 h at 600 °C.

### 4.2. Fish Experimental Design

Rainbow trout used in this experiment were an autumnal strain that reproduce during winter in France (November–December). Three-year-old males and females and two-year-old neomales were fasted for 15 days from the stop of feeding of females occurring after the reproductive event at the INRAE Lées-Athas experimental fish farm at 8 °C (Figre 8). Fifteen fish per sex (Female, Male and Neomale) were therefore sampled after the 15 days of fasting (Figure 8, Sampling 1). Remaining fish were then distributed in 18 tanks (3 replicates of tank per condition) and fed either the NC diet or the HC diet (35%, HC) for two days (5 meals). Females were fed with 9 mm pellets, while both males and neomales were fed with 5 mm pellets due to their smaller size. Fifteen fish per sex and per diet (NC and HC) were sampled after two days of feeding, 6 h after the last meal (Figure 8, Sampling 2). During sampling events (Sampling 1 and Sampling 2), fish were first anaesthetised in a benzocaine bath at 30 mg·L^−1^ and then killed by an overdose of benzocaine (at 60 mg·L^−1^). Blood was collected from the caudal vein, centrifuged at 3000× *g* for 15 min and plasma was recovered and stored at −20 °C for future plasma metabolites’ analyses. Livers were dissected into three parts and snap-frozen in liquid nitrogen to perform analysis later.

### 4.3. Plasma Metabolites

Plasma glucose and triglycerides concentrations were analysed with Glucose RTU (BioMerieux, Marcy l’Étoile, France) and PAP 150 (BioMerieux) respectively, according to the recommendations of the manufacturers.

### 4.4. RNA Extraction and cDNA Synthesis

For RNA extractions, 9 fish per sex and condition were used. Liver samples were homogenised using Precellys^®^24 (Bertin Technologies, Montigny-le-Bretonneux, France) in Trizol reagent (Invitrogen, Carlsbad, CA, USA) with 2.8 mm ceramic beads for 2 × 20 s, separated by 15 s of break, at 5500 rpm. Luciferase control RNA (Promega), 10 pg per 1.9 mg of tissue, was added to each sample to allow for data normalisation. Total RNA was then extracted according to the instructions of the manufacturer of Trizol reagent. Total RNA obtained (2 µg) was then reverse transcribed to cDNA in duplicate using the SuperScript III RNase H-Reverse Transcriptase kit (Invitrogen) with random primers (Promega, Charbonnières-les-Bains, France).

### 4.5. Quantitative Real-Time PCR

Quantitative real-time PCR was used to measure mRNA levels of genes pertinent to (1) glucose transport, with *glut2a* and *glut2b* both coding for the glucose transporter 2, (2) glycolysis, with *gcka* and *gckb* coding for the glucokinase, *pfkla* and *pfklb* coding for the phosphofructokinase and *pkl* coding for the pyruvate kinase, (3) gluconeogenesis, with *g6pca*, *g6pcb1a*, *g6pcb1b*, *g6pcb2a* and *g6pcb2b* coding for the glucose-6-phosphatase, *fbp1a*, *fbp1b1* and *fbp1b2* coding for the fructose-1,6-biphosphatase and *pck1*, *pck2a* and *pck2b* coding for the phosphoenolpyruvate kinase and (4) de novo lipogenesis, with *srebf1* coding for the sterol regulatory element-binding protein 1, *acly* coding for the ATP-citrate lyase, *aca-αa*, *aca-αb*, *aca-βa* and *aca-βb* coding for two isoforms of the acetyl-coenzyme A carboxylase, *aca-α* ohnologs coding for the isoform linked to the synthesis of fatty acid and *aca-β* ohnologs linked to the β-oxidation and *fasn* coding for the fatty acid synthase. The primer sequences used in real-time RT-PCR assays for gluconeogenic and glycolytic gene analyses were those previously described by Marandel et al. and Liu et al. Primers newly designed to study the lipogenic genes are listed in Table 5. Concerning *g6pd*, *acly* and *fasn*, primers were designed to coamplify different paralogs of the same gene. Lipogenic primers were validated using a pool of cDNA and sequencing of amplified products was performed systematically to verify the amplified sequence. Real-time RT-PCR were performed with the Roche Lightcycler 480 system (Roche Diagnostics, Neuilly-sur-Seine, France). The mix of the reaction was 6 μL per sample: 2 μL of diluted cDNA (1:76), 0.24 μL of each primer (100 mmol·L^−1^), 3 μL of Light Cycler 480 SYBR Green I Master mix and 0.52 μL of DNase/RNase-free water (5 Prime GmbH, Hamburg, Germany). The real-time RT-PCR protocol was initiated at 95 °C for 10 min for the denaturation of the cDNA and hot-start Tap-polymerase activation, followed by 45 cycles of a two-step amplification program (15 s at 95 °C; 10 s at 60 °C). Melting curves were verified systematically (temperature gradient 1.1 °C/15 s from 65 to 97 °C) at the end of the last amplification cycle to confirm the specificity of the amplification reaction. Each assay comprised replicate samples (duplicate of reverse transcription and PCR amplification, respectively) and negative controls (reverse transcriptase- and cDNA template-free samples, respectively). Data were then normalised to the exogenous luciferase transcript abundance in samples, as previously described [33].

### 4.6. Enzymatic Activities

The assessment of enzymatic activities was performed with frozen samples of liver. Samples were ground in an ice-cold buffer (50 mmol·L^−1^ TRIS, 5 mmol·L^−1^ EDTA, 2 mmol·L^−1^ DTT, protease inhibitor cocktail (Sigma, St. Louis, MO, USA)). For the fatty acid synthase (Fasn) and the glucose-6-phosphate dehydrogenase (G6pd), samples were centrifuged for 20 min at 24,000× *g* at 4 °C and supernatants were kept for the enzymatic assay. To measure pyruvate kinase (Pk) and phosphofructokinase (pfk) activities, samples were centrifuged again for 20 min at 20,000× *g* at 4 °C. For the glucose-6-phosphatase (G6pc), a sonic disruption (Bioruptor, 3 cycles, 30 s On/30 s Off), followed by a second centrifugation (10 min at 900× *g* at 4 °C) was applied, and supernatants were kept for the enzymatic assay. Enzyme activities were measured following the variation of the absorbance of nicotinamide adenine dinucleotide phosphate at 340 nm, in a Power Wave X (Biotek Instrument) reader. Reactions were started by the addition of specific substrates. Water was used as a blank for each sample. The enzymes assayed were: high km Hexokinase (Gck), as described in [23], glucose-6-phosphatase (G6pc) from [34], pyruvate kinase (Pk) following the protocol of [35], fructose-6-biphosphatase (Fbp) as described in [36], phosphofructokinase (Pfk) according to [37] and fatty acid synthase (Fasn) according to the protocol of [38].

### 4.7. Statistical Analysis

All statistical analyses were performed with the R Software (version 3.6.1). The results are presented as mean ± SD. Statistical analyses were carried out on the different parameters recorded to investigate potential differences among groups of fasted, NC and HC fish. When the normality of distributions was confirmed, one-way ANOVAs were performed on the different variables obtained. Kruskal–Wallis non-parametric tests were performed when the conditions of application of ANOVA were not respected. When significant differences were demonstrated, means were compared using a Tukey’s post hoc analysis. Statistical difference was considered significant when *p*-value was <0.05.

## 5. Conclusions

The present study investigated the glucose and lipid metabolism at the molecular level in mature female and male, and neomale, rainbow trout fed for two days with a NC diet or a HC diet. Except for the males, the results obtained for the glucose metabolism highlighted an activation of the hepatic glycolysis pathway by the nutritional status, irrespective of the diet. The upregulation of the de novo lipogenesis was also demonstrated in fed animals. These data point out that carbohydrate intake during a short period (5 meals) at low water temperature does not induce specific metabolic changes after two days of feeding in mature females and males, and neomales. Besides, we demonstrated, for the first time, sex differences regarding the consequences at the molecular level of two days of feeding, especially with neomales, displaying more pronounced regulation of the glucose metabolism than male and female broodstocks. To conclude, there is no negative effects of feeding mature female and male, and neomale, rainbow trout with a high-carbohydrate diet for a short period of time.

## Figures and Tables

**Figure 1 ijms-22-06149-f001:**
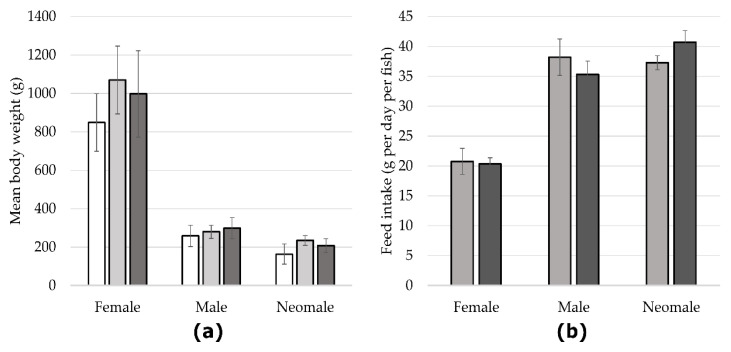
(**a**) Mean body weight and (**b**) feed intake of mature females and males and neomales fasted (white), fed with NC diet (non-carbohydrate diet, light grey) or fed with HC diet (high-carbohydrate diet, dark grey). Data (*n* = 15 fish per sex and condition) were analysed with either one-way ANOVA when the conditions of application were respected or the Kruskal-Wallis test, followed by a post hoc Tukey’s test in case of significant differences (*p* < 0.05, indicated by different letters).

**Figure 2 ijms-22-06149-f002:**
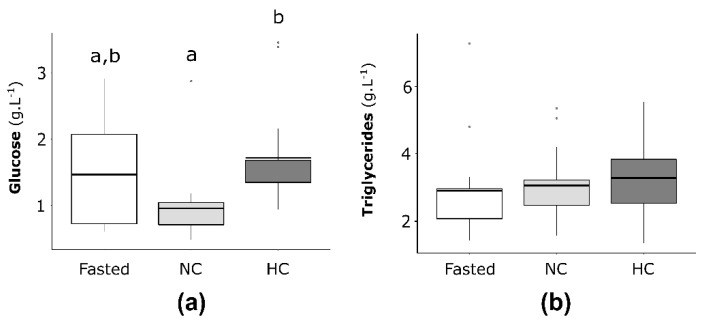
(**a**) Plasma glucose and (**b**) triglycerides concentrations of mature female rainbow trout fasted 15 days (Fasted, white), fed with NC diet (non-carbohydrate diet, grey) and fed with HC diet (high-carbohydrate diet, dark grey). Data (*n* = 15 fish per condition) were analysed with either one-way ANOVA when the conditions of application were respected or the Kruskal-Wallis test, followed by a post hoc Tukey’s test in case of significant differences (*p* < 0.05, indicated with different letters).

**Figure 3 ijms-22-06149-f003:**
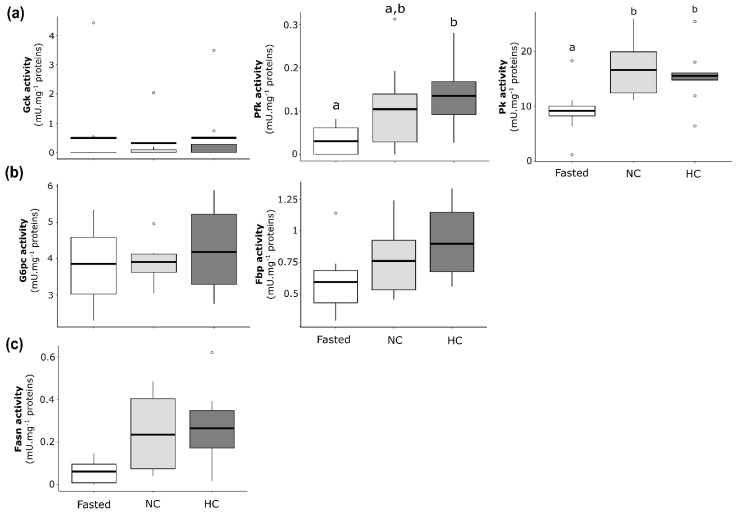
Enzymatic activities of glucose and lipid metabolism actors in the liver of mature female rainbow trout fasted (Fasted, white), fed with a non-carbohydrate diet (NC, grey) or fed with a high-carbohydrate diet (HC, dark grey). Enzymatic activities are expressed as mU·mg^−1^ of proteins for (**a**) glycolysis with the glucokinase (Gck), the phosphofructokinase (Pfk) and the pyruvate kinase (Pk), (**b**) gluconeogenesis with the glucose-6-phosphatase (G6pc) and the fructose-1,6-bisphosphatase (Fbp) and (**c**) de novo lipogenesis with the fatty acid synthase (Fasn).

**Figure 4 ijms-22-06149-f004:**
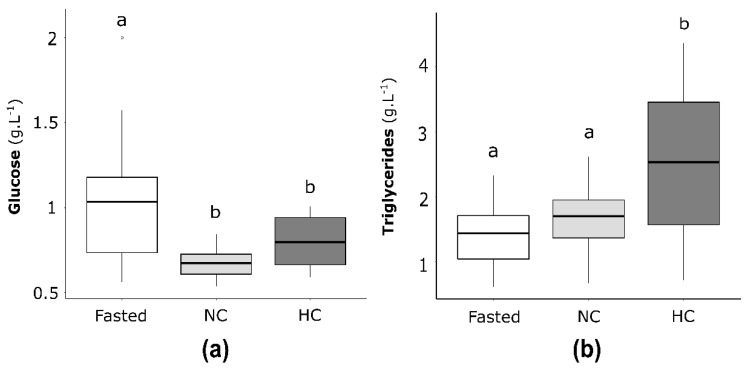
(**a**) Plasma glucose and (**b**) triglycerides concentrations of mature male rainbow trout fasted for 15 days (Fasted, white), fed with the NC diet (non-carbohydrate diet, grey) and fed with the HC diet (high-carbohydrate diet, dark grey). Data (*n* = 15 fish per condition) were analysed with either one-way ANOVA when the conditions of application were respected, or the Kruskal-Wallis test, followed by a post hoc Tukey’s test in case of significant differences (*p* < 0.05, indicated with different letters).

**Figure 5 ijms-22-06149-f005:**
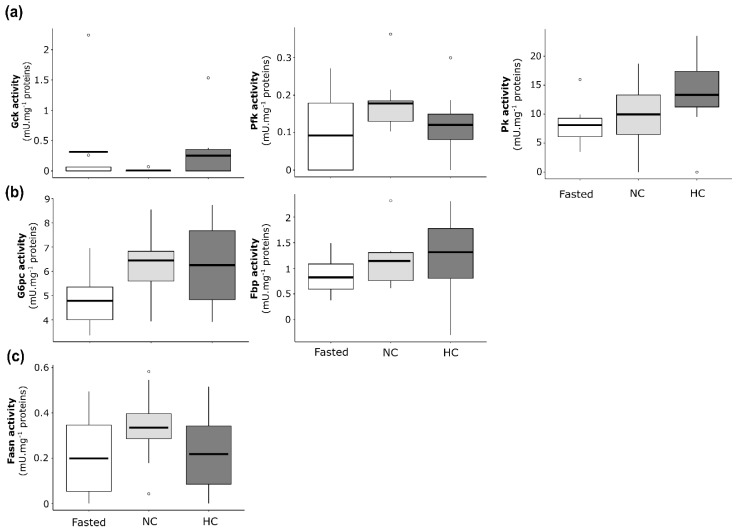
Enzymatic activities of glucose and lipid metabolism in the liver of mature male rainbow trout fasted for 15 days (Fasted, white), fed with a non-carbohydrate diet (NC, grey) and fed with a high-carbohydrate diet (HC, dark grey). Enzymatic activities are expressed as mU·mg^−1^ of proteins for (**a**) glycolysis with the glucokinase (Gck), the phosphofructokinase (Pfk) and the pyruvate kinase (Pk), (**b**) gluconeogenesis with the glucose-6-phosphatase (G6pc) and the fructose-1,6-bisphosphatase (Fbp) and (**c**) de novo lipogenesis with the fatty acid synthase (Fasn).

**Figure 6 ijms-22-06149-f006:**
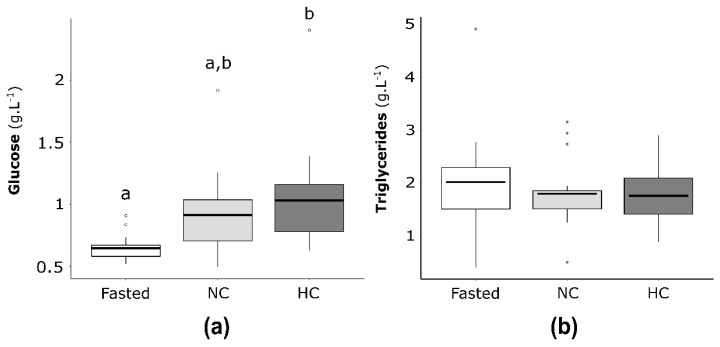
(**a**) Plasma glucose and (**b**) triglycerides concentrations of neomale rainbow trout fasted for 15 days (Fasted, white), fed with the NC diet (non-carbohydrate diet, grey) and fed with the HC diet (high-carbohydrate diet, dark grey). Data (*n* = 15 fish per condition) were analysed with either one-way ANOVA when the conditions of application were respected, or the Kruskal-Wallis test, followed by a post hoc Tukey’s test in case of significant differences (*p* < 0.05, indicated with different letters).

**Figure 7 ijms-22-06149-f007:**
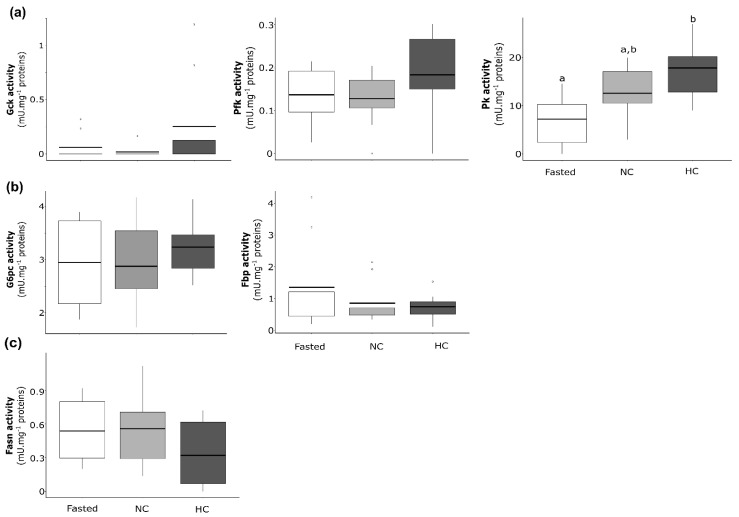
Enzymatic activities of glucose and lipid metabolism in the liver of neomale rainbow trout fasted for 15 days (Fasted, white), fed with a non-carbohydrate diet (NC, grey) and fed with a high-carbohydrate diet (HC, dark grey). Enzymatic activities are expressed as mU·mg^−1^ of proteins for (**a**) glycolysis with the glucokinase (Gck), the phosphofructokinase (Pfk) and the pyruvate kinase (Pk), (**b**) gluconeogenesis with the glucose-6-phosphatase (G6pc) and the fructose-1,6-bisphosphatase (Fbp) and (**c**) de novo lipogenesis with the fatty acid synthase (Fasn).

**Figure 8 ijms-22-06149-f008:**
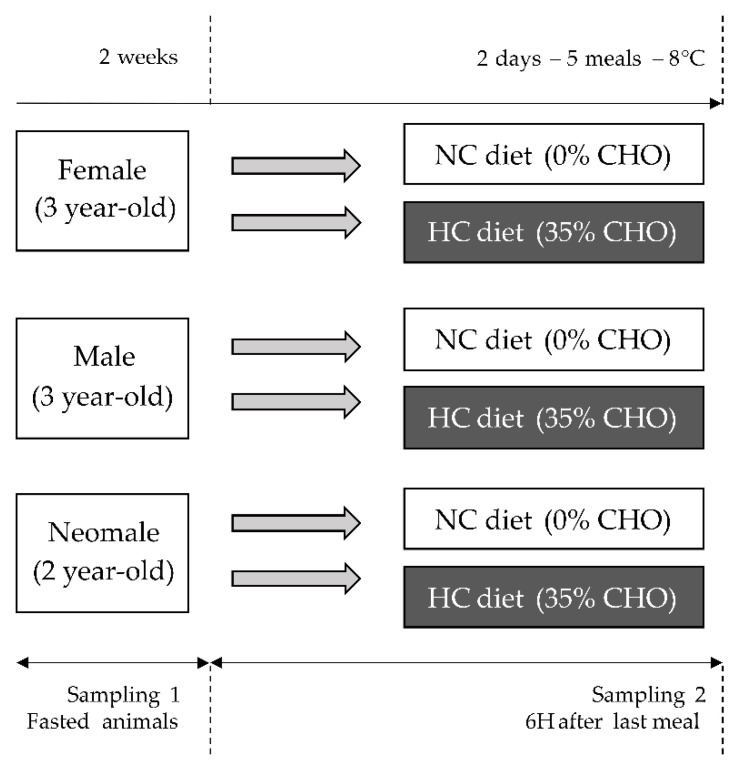
Experimental design. Mature female and male, and neomale, rainbow trout were fed either a diet containing no carbohydrates (NC) or containing 35% of carbohydrates (HC).

**Table 1 ijms-22-06149-t001:** Relative mRNA levels of glucose and lipid metabolism-related genes in liver of fasted mature female rainbow trout, fed with a non-carbohydrate diet (NC) or fed with a high-carbohydrate diet (HC). Data are represented as means ± SD (*n* = 9 females per condition). Genes which have not been detected are represented by “nd”.Data were analysed using one-way ANOVA when conditions of application were respected, otherwise the Kruskal–Wallis test was performed. Significant differences of means were analysed using post hoc Tukey’s test and are represented using different letters (^a^, ^b^) and bold *p*-values.

Pathway/Genes	Fasted	NC	HC	*p*-Value
**Glucose transport**				
*glut2a*	1.92 ± 0.60	2.36 ± 1.33	1.82 ± 1.17	0.573
*glut2b*	1.91 ± 0.64	2.37 ± 0.82	1.99 ± 1.09	0.567
**Glycolysis**				
*gcka*	3.01 ± 7.0	0	3.36 ± 5.37	0.119
*gckb*	1.83 ± 4.12 ^a,b^	0.15 ± 0.36 ^a^	4.16 ± 6.18 ^b^	**0.049**
*pfkla*	0.92 ± 0.28 ^a^	1.45 ± 0.35 ^b^	1.27 ± 0.51 ^a,b^	**0.045**
*pfklb*	0.94 ± 0.32	1.5 ± 0.57	1.3 ± 0.33	0.051
*pkl*	0.8 ± 0.35	1.37 ± 0.48	1.25 ± 0.63	0.1
**Gluconeogenesis**				
*g6pca*	0.93 ± 0.37 ^a^	0.75 ± 0.64 ^a,b^	0.42 ± 0.20 ^b^	**0.004**
*g6pcb1a*	3.08 ± 3.17 ^a^	0.94 ± 1.71 ^a,b^	0.12 ± 0.37 ^b^	**0.005**
*g6pcb1b*	0.35 ± 0.18 ^a^	0.7 ± 0.76 ^a^	0.09 ± 0.06 ^b^	**0.001**
*g6pcb2a*	0.49 ± 0.57 ^a^	1.49 ± 0.87 ^a,b^	1.8 ± 1.46 ^b^	**0.020**
*g6pcb2b*	nd	nd	nd	-
*fbp1a*	nd	nd	nd	-
*fbp1b1*	0.9 ± 0.43	1.02 ± 0.90	0.73 ± 0.23	0.848
*fbp1b2*	2.01 ± 0.67	2.11 ± 2.0	1.6 ± 0.76	0.340
*pck1*	0	0.11 ± 0.23	0	0.157
*pck2a*	1.79 ± 1.33	1.28 ± 0.58	1.45 ± 0.77	0.557
*pck2b*	1.05 ± 0.53	1.56 ± 0.72	1.47 ± 0.65	0.266
**Lipid metabolism**				
*srebf1*	0.2 ± 0.11 ^a^	0.99 ± 0.60 ^b^	1.34 ± 0.88 ^b^	**<0.001**
*g6pd*	1.63 ± 1.12	3.19 ± 2.45	2.95 ± 2.14	0.289
*acly*	0.92 ± 0.46 ^a^	2.64 ± 1.14 ^b^	2.04 ± 2.14 ^b^	**0.007**
*aca-αa*	0.15 ± 0.11 ^a^	2.08 ± 2.06 ^b^	2.72 ± 1.26 ^b^	**<0.001**
*aca-α* *b*	0.17 ± 0.09 ^a^	1.66 ± 1.49 ^b^	2.53 ± 2.44 ^b^	**<0.001**
*aca-βa*	2.53 ± 0.67	1.87 ± 2.37	2.06 ± 1.26	0.160
*aca-βb*	1.29 ± 0.59	1.3 ± 0.59	1.28 ± 0.87	0.701
*fasn*	0.12 ± 0.22 ^a^	3.28 ± 2.20 ^b^	1.67 ± 1.36 ^b^	**0.001**

**Table 2 ijms-22-06149-t002:** Relative mRNA levels of glucose and lipid metabolism-related genes in liver of fasted mature male rainbow trout (Fasted), fed with a non-carbohydrate diet (NC) or fed with a high-carbohydrate diet (HC). Data are represented as means ± SD (*n* = 9 males per condition). Genes which have not been detected are represented by “nd”. Data were analysed using one-way ANOVA when conditions of application were respected, otherwise the Kruskal–Wallis test was performed. Significant differences of means were analysed using post hoc Tukey’s test and are represented using different letters (^a^, ^b^) and bold *p*-values.

Pathway/Genes	Fasted	NC	HC	*p*-Value
**Glucose transport**				
*glut2a*	2.51 ± 1.48	2.88 ± 1.66	2.15 ± 1.21	0.790
*glut2b*	2.12 ± 1.23	2.59 ± 1.35	2.07 ± 0.96	0.656
**Glycolysis**				
*gcka*	1.61 ± 1.64 ^a^	0 ^b^	0 ^b^	**0.001**
*gckb*	0	0	0.29 ± 0.87	0.434
*pfkla*	0.6 ± 0.30	1.21 ± 0.68	0.9 ± 0.37	0.076
*pfklb*	0.59 ± 0.31	1.2 ± 0.64	0.93 ± 0.38	0.062
*pkl*	0.91 ± 0.61	1.48 ± 0.73	1.43 ± 0.39	0.100
**Gluconeogenesis**				
*g6pca*	1.85 ± 1.1	1.82 ± 1.2	1.23 ± 0.75	0.393
*g6pcb1a*	0.38 ± 0.61	0.53 ± 1.59	0	0.127
*g6pcb1b*	1.1 ± 0.7 ^a,b^	2.76 ± 3.14 ^a^	0.62 ± 0.43 ^b^	**0.046**
*g6pcb2a*	0.36 ± 0.17^a^	1.71 ± 1.1 ^b^	1.87 ± 1.5 ^b^	**0.018**
*g6pcb2b*	nd	nd	nd	-
*fbp1a*	nd	nd	nd	-
*fbp1b1*	1.66 ± 0.66	1.86 ± 0.92	1.62 ± 0.86	0.773
*fbp1b2*	0.84 ± 0.68	0.63 ± 0.43	0.74 ± 0.29	0.592
*pck1*	0.19 ± 0.23 ^a^	4.88 ± 6.35 ^b^	0.3 ± 0.3 ^a^	**0.005**
*pck2a*	2.99 ± 2.26	1.87 ± 1.09	2.50 ± 1.77	0.317
*pck2b*	1.21 ± 0.55	1.80 ± 0.83	1.33 ± 0.69	0.191
**Lipid metabolism**				
*srebf1*	0.17 ± 0.06 ^a^	1.19 ± 0.86 ^b^	0.92 ± 0.59 ^b^	**<0.001**
*g6pd*	1.64 ± 0.95	1.63 ± 1.02	1.69 ± 0.75	0.989
*acly*	0.96 ± 0.63	2.01 ± 1.27	1.57 ± 0.72	0.132
*aca-αa*	0.13 ± 0.09 ^a^	1.37 ± 1.59 ^b^	1.67 ± 1.43 ^b^	**<0.001**
*aca-αb*	0.15 ± 0.09 ^a^	1.27 ± 1.24 ^b^	1.67 ± 1.31 ^b^	**<0.001**
*aca-βa*	1.95 ± 0.84 ^a^	0.53 ± 0.39 ^b^	0.53 ± 0.36 ^b^	**<0.001**
*aca-βb*	1.18 ± 0.62	1.94 ± 1.30	1.41 ± 0.67	0.177
*fasn*	0.21 ± 0.12 ^a^	1.97 ± 1.24 ^b^	2.13 ± 1.34 ^b^	**0.004**

**Table 3 ijms-22-06149-t003:** Relative mRNA levels of glucose and lipid metabolism-related genes in liver of fasted neomale rainbow trout (Fasted), fed with a non-carbohydrate diet (NC) or fed with a high-carbohydrate diet (HC). Data are represented as means ± SD (*n* = 9 neomales per condition). Genes which have not been detected are represented by “nd”. Data were analysed using one-way ANOVA when conditions of application were respected, otherwise the Kruskal–Wallis test was performed. Significant differences of means were analysed using post hoc Tukey’s test and are represented using different letters (^a^, ^b^) and bold *p*-values.

Pathway/Genes	Fasted	NC	HC	*p*-Value
**Glucose transport**				
*glut2a*	1.96 ± 1.46	2.20 ± 1.10	1.92 ± 0.83	0.782
*glut2b*	1.64 ± 1.08	1.89 ± 1.05	1.75 ± 0.84	0.917
**Glycolysis**				
*gcka*	0	0	0.89 ± 1.51	0.125
*gckb*	0.02 ± 0.06 ^a^	0.02 ± 0.06 ^a^	2.51 ± 2.92 ^b^	**0.006**
*pfkla*	0.88 ± 0.24 ^a^	1.48 ± 0.77 ^b^	1.55 ± 0.99 ^b^	**0.042**
*pfklb*	0.83 ± 0.21 ^a^	1.43 ± 0.7 ^b^	1.46 ± 0.88 ^b^	**0.037**
*pkl*	0.75 ± 0.35	1.04 ± 0.46	1.13 ± 0.69	0.156
**Gluconeogenesis**				
*g6pca*	1.22 ± 0.37	1.04 ± 0.51	0.84 ± 0.93	0.081
*g6pcb1a*	1.92 ± 1.06a	0 b	0 b	**<0.001**
*g6pcb1b*	0.84 ± 0.74 ^a^	3.7 ± 3.25 ^b^	1.04 ± 2.10 ^a,b^	**0.015**
*g6pcb2a*	0.32 ± 0.37 ^a^	1.73 ± 1.60 ^b^	1.35 ± 1.25 ^b^	**0.001**
*g6pcb2b*	nd	nd	nd	-
*fbp1a*	nd	nd	nd	-
*fbp1b1*	0.96 ± 0.64	1.34 ± 0.67	0.76 ± 0.40	0.067
*fbp1b2*	1.3 ± 0.63 ^a^	0.97 ± 0.52 ^a,b^	0.59 ± 0.24 ^b^	**0.006**
*pck1*	0.03 ± 0.08 ^a^	5.32 ± 4.21 ^b^	2.17 ± 5.76 ^a^	**<0.001**
*pck2a*	2.99 ± 2.26	1.87 ± 1.09	2.50 ± 1.77	0.317
*pck2b*	1.21 ± 0.55	1.80 ± 0.83	1.33 ± 0.69	0.191
**Lipid metabolism**				
*srebf1*	0.4 ± 0.23 ^a^	1.2 ± 0.65 ^b^	1.9 ± 1.32 ^b^	**<0.001**
*g6pd*	0.6 ± 0.87 ^a^	2.51 ± 1.37 ^b^	1.99 ± 1.24 ^b^	**0.001**
*acly*	0.68 ± 0.9 ^a^	2.4 ± 1.08 ^b^	2.15 ± 1.53 ^b^	**0.001**
*aca-αa*	0.18 ± 0.11 ^a^	1.87 ± 1.27 ^b^	3.63 ± 3.24 ^b^	**<0.001**
*aca-αb*	0.22 ± 0.13 ^a^	1.78 ± 1.15 ^b^	3.49 ± 3.09 ^b^	**<0.001**
*aca-βa*	2.31 ± 1.23 ^a^	1.12 ± 0.74 ^a,b^	0.79 ± 0.52 ^b^	**0.005**
*aca-βb*	2.35 ± 0.60 ^a^	1.19 ± 0.52 ^b^	1.38 ± 0.78 ^a,b^	**0.005**
*fasn*	0.05 ± 0.11 ^a^	2.06 ± 1.25 ^b^	2.5 ± 1.85 ^b^	**<0.001**

**Table 4 ijms-22-06149-t004:** Diet composition of the non-carbohydrate diet (NC) and the high-carbohydrate diet (HC).

Ingredients (%)	NC		HC	
	5 mm	9 mm	5 mm	9 mm
Fish meal ^1^	77.77	77.77	45	45
Pre-gelatinised starch ^2^	-	-	37	37
CPSP90 ^3^	2	2	5	5
Soybean meal ^4^	12	12	-	-
Pea protein concentrated ^5^	-	-	5	5
Fish oil ^6^	1.66	1.66	3.96	3.96
Cellulosis ^7^	2.53	2.53	-	-
Alginate ^8^	2	2	2	2
Minerals premix ^9^	1	1	1	1
Vitamins premix ^9^	1	1	1	1
Astaxanthine ^10^	0.04	0.04	0.04	0.04
**Proximate composition**				
Dry matter	93.97	94.42	95.84	96.05
Crude proteins (%DM)	66.63	65.7	41.97	42.94
Gross energy (kJ/gDM)	20.21	20.48	18.73	20.44
Ash (%DM)	16.81	16.75	10.37	10.4
Carbohydrates (%DM)	0.55	0.55	32.55	32.55
Crude lipids (%DM)	8.71	9.69	7.57	6.47

^1^ Sopropêche 62126 Wimille, France, ^2^ Gelatinised corn starch roquette 62136 Wimille, France, ^3^ Sopropêche 62126 Wimille, France, ^4^ SUDOUEST Aliment 40360 Pomarez, France, ^5^ roquette 62136 Lestrem, ^6^ Sopropêche 62126 Wimille, France, ^7^ RETTENMAIER FRANCE SARL 78100 Saint Germain en Laye, France, ^8^ Louis François 77,134 Croissy Beaubourg, France, ^9^ INRAE SAAJ 78350 Jouy en Josas, France, ^10^ DSM food 59113 Seclin, France.

**Table 5 ijms-22-06149-t005:** Primer sequences and accession numbers for qPCR analysis of the de novo lipogenesis pathway.

Gene	Forward Primer (5’-3’)	Reverse Primer (5’-3’)	GenBank Accession Number
*srebf1*	CAGTTGCTGCTGTGTGACCT	TGATGTGTTCGTGTGGGACT	XM_021624594.1
*acly*	CCTCTGTAAGGCCAAGTGGG	TTGGCATCCAGGTCTGCAAT	XM_021624819.1/XM_021557697.1/XM_021575392.1
*aca-α* *a*	ACAGGACCCTAAAGCACAGG	GGTGAAAGAGGTGTCCAGGA	XM_021623125.1
*aca-α* *b*	TCCAGTTCATGCTGCCTACC	GCTTAATGTCCCGAGTGCGA	XM_021618451.1
*aca-β* *a*	TCGCTCAGAATTCCGGGTAC	CGCGTGGTGATGGTTACAAT	XM_021605386.1
*aca-β* *b*	TGAACAGCTTGGTAAACAGCC	TCTCGTGCATTCTACCAGGG	XM_021620987.1
*fasn*	TGATCTGAAAGCCCGTGTCAA	ATTTGGTTGCCGGGACGTAA	XM_021576228.1/XM_021581290.1

## Data Availability

Data are available by simple request from Lucie Marandel.

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
