# Peer review of "Short-Term Effect of a Low-Protein High-Carbohydrate Diet on Mature Female and Male, and Neomale Rainbow Trout"

_ijms, 2021, doi:10.3390/ijms22116149_

Round 1
Reviewer 1 Report
The Ms by Favalier et al deals with a short term study performed on males, females and neomale rainbow trout fed on a low protein / high carbohydrate diet. The research group perfeomes different analyses to better unserdstand lipid and lucose metabolism.
The Ms is well written and results are clear.
I only have a few minor requests for the authors.
Please better explain why you did not test a longer period of time during the experiment.
Introduction .
A short description on how the diets for trout changed over the past years is necessary. Many of the commercial diets are not based on FM but on vegetable proteins or new ingredients.
For your convenience you can read or cite the present MS:
Physiological response of rainbow trout (Oncorhynchus mykiss) to graded levels of Hermetia illucens or poultry by-product meals as single or combined substitute ingredients to dietary plant proteins
Randazzo, B., Zarantoniello, M., Gioacchini, G., ...Tibaldi, E., Olivotto, I. Aquaculture, , ,Dietary inclusion of full-fat Hermetia illucens prepupae meal in practical diets for rainbow trout (Oncorhynchus mykiss): Lipid metabolism and fillet quality investigations
Bruni, L., et al. Aquaculture, , ,
Author Response
Point 1: Please better explain why you did not test a longer period of time during the experiment
Response 1: We did not test a longer period of time because our main objective was to assess the direct metabolic consequences of feeding broodstock with a high carbohydrate diet after a fasting period on their glucose and lipid metabolism and how the metabolism is directly affected after the refeeding period at the beginning of the reproductive cycle (i.e. right after spawning for female).
Point 2: A short description on how the diets for trout changed over the past years is necessary. Many of the commercial diets are not based on FM but on vegetable proteins or new ingredients.
Response 2: We thank the reviewer for his suggestion and thus added to the manuscript information regarding the evolution of diet formulation (lines 24-27)
Point 3: Please provide extra details on feeding habits of females during reproduction
Response 3: We provided more information regarding the feeding habits of females (lines 327-330)
Point 4: A link with egg development and vitellogenesis may be interesting for the readers
Response 4: We agree that egg development and vitellogenesis are interesting regarding the influence of the composition of diet of females however we sampled fish and assess the consequences of feeding with a high carbohydrate after the reproduction and the refeeding of fish, fish were fed before their reproduction with commercial diet and consecutively fast for 15 days. Callet et al., (2020) who highlighted changes in carbohydrate metabolism during gametogenesis, discussed these thematic. Moreover, fish in our facility were fed for several months after the sampling for this article and the results obtained will be the subject of a separate study
Point 5: I also suggest the authors, in order to improve the discussion to read this other MS.
Response 5: We thank the reviewer for this suggestion and we read the manuscript but unfortunately, we did not find, regarding the specificity of our study, any relevant additional feature that we could add in our manuscript
Point 6: Please provide in the ethical statement the specific authorization number of your experiment
Response 6: The ethical committee concluded that regarding our experimentation (two days of feeding) and the fact that this study follow the rules stated by the French guiding principles for the use and care of laboratory animals, this study did not require a specific ethical statement.
Reviewer 2 Report
The manuscript entitled 'Short-term effect of a low protein high carbohydrate diet on mature females and males, and neomales rainbow trout' is clearly described.
The Introduction section describes the topic in detail, contains the information necessary for a general overview of the study and provides references in line with the content of the paper.
In the Materials and Methods section, the methods and treatments performed are explained in detail.
The results are presented in a logical sequence in the text, are easy to understand and respect the objective of the work.
The discussion is extensive, well described and supported by bibliographical studies.
Author Response
We thank reviewer 2 for his review.